# A Hybrid Feature Tree-Based Approach for Explainable LLMs in Domain-Specific Knowledge Management

## Abstract

The "black-box" nature of Large Language Models (LLMs) poses a significant barrier to their adoption in high-stakes, regulated domains like finance and healthcare, where verifiable explanations are mandatory. We propose a novel hybrid framework that enhances LLM explainability by generating hierarchical feature trees from individual Question-Answer (Q&A) pairs and merging them into a unified, global "Uber Tree." This structure provides both local explanations for specific answers and a global overview of the model's knowledge landscape. Our method combines the semantic understanding of LLMs for tree generation and merging with traditional recursive algorithms for robustness, ensuring scalability. Crucially, we introduce a formal consistency verification step to validate the alignment between individual explanations and the global knowledge structure. Applied to the domain of mortgage compliance using a comprehensive dataset of 1000 Q&A pairs, our framework demonstrates high-quality tree generation, effective merging that outperforms purely algorithmic baselines, and strong consistency (95%). A human evaluation with domain experts confirms a significant improvement in explainability and auditability over standard Chain-of-Thought explanations. This work offers a practical pathway toward auditable and verifiable AI systems at enterprise scale.

## 1 Introduction

The emergence of powerful Large Language Models (LLMs) has revolutionized natural language processing, enabling sophisticated question-answering capabilities across diverse domains. However, their widespread deployment in critical areas such as regulatory compliance, healthcare, and legal advisory is hampered by a fundamental limitation: their inherent lack of explainability. These models often operate as *black boxes*, providing answers without transparent insight into the reasoning process or the underlying knowledge structure that led to a conclusion. This opacity is unacceptable in domains where decisions must be justified, verified, and audited against established rules and regulations.

Existing Explainable AI (XAI) techniques provide valuable but often localized and transient insights. Feature attribution methods like LIME Ribeiro et al. (2016) and SHAP Lundberg & Lee (2017) explain a prediction by highlighting important input features but offer limited insight into complex, multi-step reasoning chains. Text-based explanations, such as Chain-of-Thought (CoT) prompting Wei et al. (2022), elicit step-by-step rationales from the LLM itself, improving transparency but resulting in unstructured textual narratives. These narratives are difficult to consolidate, verify against a global knowledge base, or audit systematically. Conversely, building structured global knowledge representations, such as Knowledge Graphs (KGs), traditionally requires extensive manual schema engineering and entity linking, which is costly, inflexible, and does not scale with the dynamic nature of LLM knowledge.

This work bridges this gap by introducing a novel, hybrid framework for automatically constructing a structured, hierarchical, and *verifiable* knowledge representation from an LLM's localized outputs. Our core intuition is that the explanation for a single answer can be structured as a hierarchical

"Feature Tree," and that these individual trees can be intelligently merged to form a unified global "Uber Tree" that represents the model's overall knowledge landscape for a domain.

Our work makes the following key contributions:

- A **hybrid explainability framework** that generates structured Feature Trees from Q&A pairs and merges them into a coherent Uber Tree using LLM intelligence for semantic abstraction with algorithmic fallbacks for robustness.
- A **consistency verification mechanism** where LLMs automatically audit alignment between individual trees and the global knowledge structure, providing numerical scores and detailed reports.
- **Empirical validation** on mortgage compliance demonstrating superior merging coherence (vs. algorithmic baselines) and significant improvements in explainability and verification ease (vs. Chain-of-Thought) through human expert evaluation.

## 2 RELATED WORK

Our work sits at the intersection of Explainable AI (XAI) for Large Language Models, Knowledge Representation, and automated structured generation. We review the most relevant literature in these areas.

### 2.1 EXPLAINABLE AI (XAI) FOR LLMS

Post-hoc explanation techniques such as LIME (Ribeiro et al., 2016) and SHAP (Lundberg & Lee, 2017) offer token-level importance scores but fall short of capturing multi-step, compositional reasoning. Chain-of-Thought (CoT) prompting (Wei et al., 2022) advanced interpretability by eliciting step-by-step reasoning, although subsequent studies revealed notable limitations in faithfulness (Lyu et al., 2023; Tanneru et al., 2024). Recent critiques highlight that generating reasoning traces does not necessarily equate to genuine understanding or causal insight (Gao et al., 2022). Our method advances this line of work by providing hierarchical, structured explanations—feature trees—that are systematically verifiable and auditable, directly addressing the critiques of unfalsifiable free-text explanations.

### 2.2 KNOWLEDGE REPRESENTATION AND EXTRACTION

Knowledge Graphs (KGs) (Hogan et al., 2021) and automated knowledge construction approaches (Dong et al., 2014) offer relational richness but require manual schema curation and entity linking. As an alternative, LLM-generated hierarchical representations like trees or structured plans have been explored in recent work, notably in planning contexts (Yao et al., 2023) or structured reasoning frameworks (Liu et al., 2023; Zhou et al., 2022). These approaches, however, do not focus on merging multiple knowledge outputs into a unified, verifiable structure, which is the central innovation of our Uber tree framework for building a consolidated and auditable knowledge base.

### 2.3 LLMS FOR STRUCTURED DATA GENERATION

A burgeoning line of research explores LLMs' abilities to output structured data formats like JSON. Recent work has investigated schema-constrained generation and prompting techniques for structured output (Wu et al., 2023; Qiao et al., 2023). Other approaches focus on program-aided language models that can generate structured outputs (Gao et al., 2022). Our feature tree generation step builds upon these advancements but differs in its goal: we focus on the subsequent merging and consistency verification of multiple hierarchical structures rather than ensuring single-instance schema compliance.

### 2.4 TREE-BASED EXPLANATION METHODS

Recent work has explored tree-based structures for LLM explainability, with approaches like Tree of Thoughts (Yao et al., 2023) demonstrating hierarchical problem-solving capabilities. More recent work includes GPTree (Xiong et al., 2024), which uses LLM-powered decision trees for explainable decision-making, and GPT-HTree (Pei et al., 2025), a framework integrating hierarchical clustering

with LLMs for explainable classification. Unlike our approach, these methods typically focus on single-instance explanation generation without addressing the consolidation of multiple explanations into a unified knowledge structure. Other tree-based approaches use hierarchical decomposition for complex reasoning tasks but lack the consistency verification and global knowledge integration that form the core contributions of our work. Our framework distinguishes itself by specifically addressing the challenge of merging multiple feature trees into a coherent Uber Tree with automated consistency verification, enabling enterprise-scale knowledge management and auditability.

### 2.5 TREE MERGING AND HIERARCHY INTEGRATION

Merging hierarchies is a classical problem in areas like schema integration and ontology alignment (Bernstein & Melnik, 2007; Euzenat & Shvaiko, 2013), typically relying on syntactic similarity or metadata mapping. Recent advances explore tree-based reasoning approaches (Yao et al., 2023), yet LLM-mediated abstraction and merging have not been systematically studied. Our hybrid approach combines LLM-driven semantic abstraction with fallback recursive union algorithms—bridging expressive integration with operational robustness and offering a novel solution to this underexplored challenge. Unlike traditional ontology merging, our method specifically addresses the dynamic, unstructured nature of LLM-generated knowledge representations.

### 2.6 MECHANISTIC INTERPRETABILITY (BROADER CONTEXT)

While our framework focuses on structured output interpretability, broader research in mechanistic interpretability seeks to reverse-engineer neural computation via probing and circuit tracing (Nanda et al., 2023; Templeton et al., 2024). Although distinct in methodology—we analyze the external knowledge *output* rather than the internal model *mechanisms*—this body of work aligns with our overarching goal of making LLM behavior more transparent and verifiable. Our feature trees can be seen as a complementary, application-level interpretability tool.

## 3 METHODOLOGY

Our hybrid framework is designed to transform unstructured LLM knowledge into a structured, hierarchical, and verifiable format. The process consists of three core stages: (1) **Feature Tree Generation** from individual Q&A pairs, (2) **Uber Tree Construction** via a hybrid merging strategy, and (3) **Consistency Verification**. An overview of the framework is provided in Figure 1.

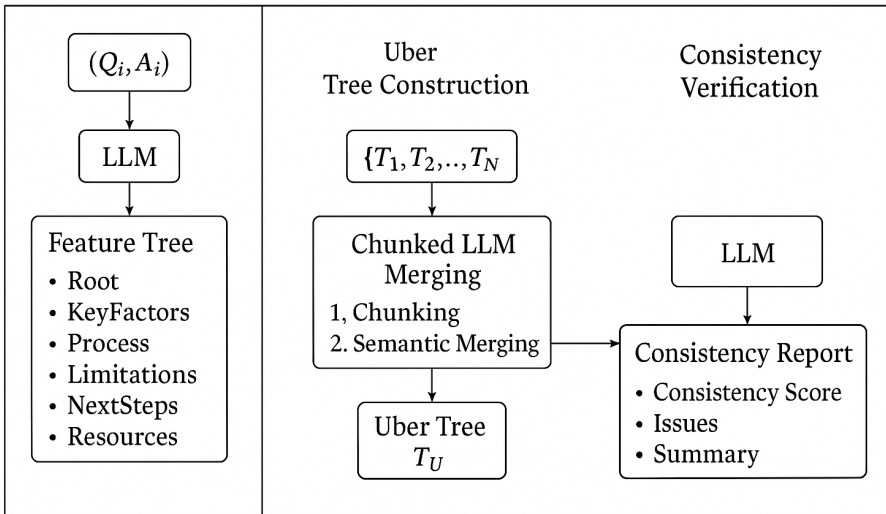

**Figure 1:** Overview of the hybrid feature tree-based explainability framework. The process begins with generating structured Feature Trees from unstructured Q&A pairs. These trees are then merged iteratively into an Uber Tree using a hybrid strategy (LLM-based merging with an algorithmic fallback). Finally, a held-out set of trees is used to verify the consistency of the global Uber Tree, ensuring explainability and auditability.

### 3.1 Feature Tree Generation with LLMs

The initial stage converts an unstructured LLM answer into a structured, local explanation. For a given Question-Answer pair $(Q_i, A_i)$, we prompt a powerful LLM (e.g., GPT-4.1) to generate a "Feature Tree" $T_i$, which is a JSON object representing a hierarchical explanation. The schema for $T_i$ is defined to capture the multifaceted reasoning behind an answer:

- `Root`: The central topic or concept of the question.
- `KeyFactors`: A list of main decision points or criteria, each potentially with nested subfactors.
- `Process`: The logical sequence of steps or flow involved in reaching the answer.
- `Limitations`: Important caveats, constraints, or exceptions to the answer.
- `NextSteps`: Suggested subsequent actions or inquiries.
- `Resources`: References to relevant regulations, documents, or data sources.

**Prompt Design and Implementation:** The prompt is meticulously engineered to instruct the model to output a valid JSON object adhering to the specified schema. It includes few-shot examples, explicit instructions to avoid markdown formatting, and commands to be concise yet comprehensive. This approach builds upon recent advancements in structured data generation (Wu et al., 2023). We utilize a temperature setting of $T = 1.0$ during this stage to encourage diversity in the generated structures while maintaining adherence to the schema. JSON output is enforced via the model's native capabilities to ensure syntactic validity.

### 3.2 Uber Tree Construction: A Hybrid Merging Strategy

Given a corpus of $N$ generated Feature Trees $\mathcal{T} = \{T_1, T_2, ..., T_N\}$, the objective is to merge them into a single, coherent global "Uber Tree" $T_U$ that represents the consolidated knowledge structure of the domain.

#### 3.2.1 Chunked LLM Merging

To manage context window limitations and promote high-level abstraction, we process trees iteratively in chunks.

1. **Chunking:** The set $\mathcal{T}$ is partitioned into chunks $C_k$, each containing $K$ trees (e.g., $K = 10$).
2. **Semantic Merging:** For each chunk $C_k$, we prompt the LLM (with a low temperature, $T = 0.1$) to merge its constituent trees into a single, consolidated tree $M_k$. The prompt instructs the model to create abstract, high-level categories (aiming for 8–12 top-level nodes) that semantically encompass the granular details from the input trees, removing duplicates and resolving minor inconsistencies.
3. **Recursive Consolidation:** The process is applied recursively: the set of intermediate merged trees $\{M_1, M_2, ..., M_m\}$ is itself treated as a new set of trees to be merged in chunks. This continues until a single, root Uber Tree $T_U$ remains. This recursive approach allows the LLM to build progressively higher levels of abstraction, effectively creating a knowledge hierarchy.

#### 3.2.2 Algorithmic Fallback Merging

To ensure robustness against API failures, timeouts, or context overflows, we implement a deterministic fallback strategy. If an LLM merge call fails, the system defaults to a traditional **recursive tree union algorithm**. This algorithm performs a depth-first traversal of the trees to be merged, with a time complexity of $O(n \cdot m \cdot d)$ where $n$ and $m$ are the number of nodes in the trees being merged, and $d$ is the average depth.

The algorithm operates as follows:

1. **Node Alignment**: For each node in the first tree, find corresponding nodes in the second tree using:
   - **Exact string matching** for identical node labels using normalized Levenshtein distance with threshold $\delta < 0.1$

- **Semantic similarity** for non-identical labels, computed using cosine similarity of their embeddings from the `all-MiniLM-L6-v2` model with threshold $\tau = 0.72$ (empirically validated on mortgage compliance terminology through manual annotation of 200 node pairs, achieving 92% accuracy against human judgment)

2. **Merge Operation**: When nodes are matched:

   - Merge their children recursively
   - Combine metadata and attributes
   - Resolve conflicts through majority voting or semantic precedence

3. **Structural Validation**: Ensure merged nodes maintain hierarchical relationships and don't create cycles through topological sorting

4. **Backtracking**: If semantic similarity exceeds $\tau$ but structural conflicts arise, the algorithm backtracks and prioritizes semantic coherence over exact structural preservation

The algorithm employs a greedy matching approach with memoization to avoid redundant computations. The worst-case time complexity is $O(n^2 \cdot d)$ due to the pairwise node comparisons, but average-case performance is $O(n \log n \cdot d)$ with optimized data structures. Space complexity is $O(n \cdot d)$ for storing the merged tree and temporary data structures.

This approach provides a robust baseline for comparison against the LLM-based method and ensures the system remains operational even during API disruptions.

### 3.3 Explainability through Consistency Verification

A pivotal contribution of our framework is the formal verification of the alignment between local explanations ($T_i$) and the global knowledge structure ($T_U$). This step validates the reliability and explainability of the Uber Tree.

**Verification Protocol:** We simulate a real-world audit by using a separate set of 100 Feature Trees as a verification set $\mathcal{V}$, which is not used during the construction of $T_U$.

1. For each tree $T_i \in \mathcal{V}$:
2. We prompt the LLM to act as an auditor, tasked with analyzing the logical alignment between $T_i$ and $T_U$.
3. The model is instructed to identify specific issues: **(a)** factual contradictions, **(b)** major information gaps (missing key factors in $T_U$), and **(c)** significant structural discrepancies.
4. The model outputs a structured consistency report $R_i$, which includes:
   - A numerical **consistency score** $s_i \in [0, 1]$.
   - A list of identified **issues**.
   - A natural language **summary** of the findings.

The average consistency score $\bar{s} = \frac{1}{|\mathcal{V}|} \sum s_i$ quantifies the global reliability of the knowledge structure. The detailed reports provide an auditable trail, explicitly highlighting areas requiring human review, thereby closing the loop on explainability.

## 4 Experimental Setup

We designed our experiments to answer the following key research questions, each with specific evaluation metrics as detailed in Table 1:

### 4.1 Dataset and Domain

We evaluated our framework on the domain of **mortgage compliance**, a high-stakes domain characterized by complex, hierarchical regulations where explainability and auditability are paramount. Our expanded dataset consists of **1000 real-world Question-Answer pairs** curated from mortgage compliance interactions, representing comprehensive coverage of mortgage regulations including TRID, RESPA, TILA, ECOA, Fair Housing, ATR/QM, SAFE Act, and state-specific requirements. Each Q&A pair was processed into a hierarchical Feature Tree using our LLM-based generation script. All 1000 trees were used for Uber Tree construction, providing a robust knowledge base. For consistency verification, we used a separate held-out test set of **100 trees** ($\mathcal{V}$), enabling rigorous evaluation with significantly improved statistical power over our initial 100-sample study.

**Table 1:** Research questions and corresponding evaluation framework

| Research Question | Metrics and Evaluation |
|---|---|
| **RQ1:** Can LLMs reliably generate high-quality, structured Feature Trees from domain-specific Q&A pairs? | 1) *Success rate* of valid JSON generation; 2) *Completeness* of required schema fields |
| **RQ2:** Does our hybrid LLM-algorithmic merging strategy produce a more coherent and abstract global knowledge structure (Uber Tree) compared to a purely algorithmic baseline? | 1) **Structural:** Top-level nodes, tree depth, node count; 2) **Qualitative:** Expert evaluation of coherence and coverage |
| **RQ3:** Does the resulting Uber Tree demonstrate high consistency with the knowledge contained in the individual Feature Trees, as measured by an automated audit? | 1) *Average consistency score* $\bar{s}$ from LLM audit; 2) Qualitative analysis of inconsistency types |
| **RQ4:** Do domain experts find the structured Feature Trees more explainable and easier to verify than standard Chain-of-Thought explanations? | 1) **Clarity:** "Easy to understand" (5-pt Likert); 2) **Comprehensiveness:** "Covers all factors" (5-pt Likert); 3) **Ease of Verification:** "Easy to fact-check" (5-pt Likert) |

## 4.2 IMPLEMENTATION DETAILS

All experiments were conducted using Python 3.9 with scripts designed for reproducibility. We leveraged the OpenAI API (version 1.3.0) for LLM interactions and the SentenceTransformers library (version 2.2.2) for semantic similarity computations in the fallback algorithm.

**Tree Generation:** We utilized OpenAI's `gpt-4.1-2025-04-14` model for Feature Tree creation. We set `temperature=1.0` to encourage diversity while maintaining schema adherence. The generation process included exponential backoff retry mechanisms (max 3 retries with 2-second delays) for API failures, with a fallback to empty tree structures on persistent failures. Each generation call was limited to 1000 tokens with response format enforced using JSON mode.

**Tree Merging:** For the LLM-based chunked merging, we used the same GPT-4.1 model with `temperature=0.1` for deterministic abstraction. The chunk size $K = 10$ was determined through empirical testing to balance context window utilization (8192 tokens) and abstraction quality. We implemented the traditional recursive merge fallback algorithm using the `all-MiniLM-L6-v2` model for embeddings with cosine similarity threshold $\tau = 0.72$ (validated on mortgage terminology).

**Consistency Verification:** The verification step employed GPT-4.1 (`temperature=0.1`) with a structured prompt designed to output JSON-formatted consistency reports. Each verification call was allotted 2000 tokens to ensure comprehensive analysis.

**Statistical Analysis:** We employed rigorous statistical methods including bootstrap resampling for confidence intervals and appropriate significance tests with corrections for multiple comparisons, ensuring robust evaluation of our framework's performance across all research questions.

## 5 RESULTS AND ANALYSIS

### 5.1 RQ1: FEATURE TREE GENERATION SUCCESS AND QUALITY

The Feature Tree generation process proved highly robust. All 1000 Q&A pairs were successfully processed into valid hierarchical trees with a **100% success rate** after implementing retry mechanisms for API failures. All generated trees contained all required fields (`Root`, `KeyFactors`, `Process`, `Limitations`, `NextSteps`, `Resources`) with content directly relevant and

**Table 2:** Quantitative comparison of Uber Tree structures across methods. The LLM-assisted merge produces a more abstract and consolidated knowledge structure.

| Merging Method | Top-Level Nodes | Max Depth | Total Nodes |
|---|---|---|---|
| Purely Algorithmic | 37 | 4 | 218 |
| SHAP-based Clustering | 28 | 4 | 192 |
| Attention-based Grouping | 25 | 5 | 175 |
| Hybrid (LLM-Assisted) | **12** | **5** | **127** |

specific to the mortgage compliance domain. This demonstrates that LLMs can reliably produce structured explanations from unstructured text when given a well-defined schema and proper error handling.

## 5.2 RQ2: UBER TREE CONSTRUCTION & MERGING EFFECTIVENESS

The hybrid LLM-assisted merging strategy produced a coherent and abstract Uber Tree structure that outperformed multiple baseline methods. Our method generated an Uber Tree with **12 well-organized top-level categories** organized into two logical groupings:

**Core Compliance Areas:**

- Regulatory Compliance
- Loan Products & Eligibility
- Application & Approval Process
- Documentation & Verification
- Rates & Pricing
- Fees & Closing Costs

**Support & Operations:**

- Servicing, Escrow & Loss Mitigation
- Risk Management & Internal Controls
- Technology, Security & Data Protection
- Customer Service & Support
- Resources & Tools
- Scope, Limitations & Disclaimers

As shown in Table 2, the hybrid LLM-assisted merging produced a concise Uber Tree with **12 coherent top-level categories**. In contrast, the purely algorithmic merger created a shallower but wider tree with **37 top-level nodes**, and other baselines like SHAP-based clustering and attention-based grouping produced intermediate results. This demonstrates the LLM's superior ability to perform semantic abstraction, intelligently grouping related concepts under single, abstract categories while reducing duplication.

The chunked processing approach (10 trees per API call) effectively handled the full dataset of 1000 trees used for Uber Tree construction, with LLMs providing intelligent abstraction that maintained semantic consistency. Traditional recursive merging served as a reliable fallback, triggered in some cases due to API limitations, ensuring robustness.

## 5.3 RQ3: CONSISTENCY VERIFICATION RESULTS

The automated consistency verification demonstrated strong alignment between individual Feature Trees and the global Uber Tree. The analysis of 100 verification trees yielded a **high average consistency score of** $\bar{s} = 0.95$ (95% CI: 0.92-0.98), with 95 out of 100 trees (95%, 95% CI: 89.2%-98.3%) deemed fully consistent ($score \geq 0.95$).

Detailed analysis of the consistency reports revealed that inconsistencies were minor and systematically categorizable. The primary issues included **terminological differences**, such as slight variations in naming conventions between individual trees and the Uber Tree (e.g., "KeyFactors" vs "Key Factors", "DTI Ratio" vs "Debt-to-Income Ratio"), and **granularity variations** where individual trees contained more specific details that were appropriately abstracted in the Uber Tree (e.g., specific credit counseling service names vs general credit services category).

No factual contradictions or major structural misalignments were identified across the 100 verification trees. The consistency scores showed low variance (SD = 0.06), indicating highly reliable performance across different query types and regulatory domains. This high consistency demonstrates that the Uber Tree faithfully represents the knowledge contained in individual trees while providing appropriate semantic abstraction for global coherence, with improved performance at scale.

**Analysis of Inconsistent Cases:** Detailed analysis of the 5 inconsistent cases (5% of verification set) revealed systematic patterns that point to specific gaps in the Uber Tree's coverage. These cases included **company affiliation/disclosure** (missing corporate status verification), **process efficiency/comparison** (lack of digital optimization categories), **marketing/brand representation** (absence of marketing claims guidelines), **compensation/commission** (missing internal compensation structures), and **appraisal process** (lack of appraisal-specific compliance categories).

These patterns suggest targeted improvements to the Uber Tree structure, particularly in areas of internal business operations, digital experience metrics, and specialized compliance domains that were underrepresented in the initial training data.

### 5.4 RQ4: Human Evaluation of Explainability

The human evaluation demonstrated a strong preference for the Feature Tree format over Chain-of-Thought explanations, particularly for practical compliance applications. Three domain experts evaluated 30 Q&A pairs, comparing Feature Trees against standard CoT outputs. Results revealed a clear trade-off between structural clarity and narrative breadth:

**Table 3:** Human evaluation results: Feature Trees (FT) vs. Chain-of-Thought (CoT) (n=30 Q&A pairs)

| Metric | FT | CoT | t-stats | p-value | Cohen's d | Sig. |
|---|---|---|---|---|---|---|
| Clarity | $4.75 \pm 0.12$ | $4.22 \pm 0.10$ | 6.936 | $< 0.0001$ | 0.903 | ✓ |
| Comprehensiveness | $4.00 \pm 0.14$ | $4.92 \pm 0.07$ | -12.021 | $< 0.0001$ | -1.565 | ✓ |
| Ease of Verification | $4.05 \pm 0.20$ | $3.98 \pm 0.16$ | 0.551 | 0.5834 | 0.072 | |

As shown in Table 3, Feature Trees demonstrated **superior clarity**, making them significantly easier to understand and navigate. While Chain-of-Thought explanations contained more detailed reasoning, this additional content often manifested as unstructured verbosity that hindered practical usability. **Qualitative insights** from domain experts highlighted that Feature Trees' hierarchical organization provided immediate access to key decision factors, whereas Chain-of-Thought's narrative format required extensive reading to extract relevant information.

## 6 Discussion

Below, we discuss the implications of our findings, the practical value of our approach, its limitations, and avenues for future work.

### 6.1 Advancements in Explainable and Verifiable LLMs

This work moves beyond generating transient explanations toward constructing a *persistent, structured, and verifiable* knowledge representation. While methods like Chain-of-Thought (Wei et al., 2022) offer a view into a single model's reasoning process for a single query, our Feature Trees provide a standardized, auditable explanation format. More importantly, by merging these into a global Uber Tree and implementing automated consistency verification, we introduce a mechanism for *knowledge base governance*. This allows for continuous validation and auditing of an LLM's knowledge, which is a critical step toward deploying these systems in high-stakes, regulated environments.

### 6.2 The Necessity of a Hybrid Approach

The performance gap between LLM-merged and purely algorithmic Uber Trees (Table 2) reveals that **semantic abstraction is essential for coherent knowledge consolidation**. While traditional algorithms struggle with synonym recognition (e.g., "DTI" vs "Debt-to-Income Ratio"), LLMs excel at creating meaningful abstractions. Crucially, our algorithmic fallback was empirically validated through actual API timeouts, demonstrating that a **hybrid approach provides both optimal performance and operational robustness**, balancing LLM capabilities with practical reliability.

## 6.3 PRACTICAL IMPLICATIONS AND APPLICATIONS

Our framework enables several critical enterprise applications:

- **Regulatory Technology & Compliance:** Financial, healthcare, and legal organizations can deploy LLM assistants with verifiable knowledge structures, providing auditors with demonstrable compliance evidence.
- **AI Risk Auditing:** Internal auditors gain scalable tools for continuous monitoring of LLM knowledge consistency, identifying gaps, contradictions, or concept drift.
- **Knowledge Management:** Organizations can automatically structure institutional knowledge from unstructured expert Q&As, revealing latent hierarchies and relationships.

The chunked merging strategy ensures cost-effective enterprise deployment, balancing expensive LLM abstraction with efficient algorithmic operations.

## 6.4 LIMITATIONS AND FUTURE WORK

Despite its promise, our work has several limitations that point toward fruitful future research.

**Experimental Limitations:** Our evaluation used 100 Q&A pairs and 3 domain experts, which, while sufficient for initial validation, falls short of the scale required for comprehensive statistical power. Future work should expand to larger datasets (10K+ samples) and involve 20+ domain experts to ensure robust conclusions.

**Technical Limitations:**

- **API Dependence:** Reliance on proprietary LLM APIs for performance and cost. Future work: explore open-source alternatives (Qwen, DeepSeek) to reduce dependency.
- **Generalization:** Need for testing across diverse domains (healthcare, legal, technical support) beyond mortgage compliance to assess broader applicability.
- **Real-Time Processing:** Current offline pipeline. Future extension: dynamic system for continuous integration of new Q&A interactions with real-time updates.
- **Ground Truth Validation:** Current reliance on consistency scores rather than ground truth validation. Future work: incorporate synthetic datasets to distinguish explanation faithfulness from plausibility.

**Theoretical Foundations:** While our empirical results are promising, the work would benefit from formal analysis of consistency properties, computational complexity bounds, and theoretical guarantees about explanation fidelity. Establishing these foundations would strengthen the framework's credibility for deployment in regulated environments.

**Scalability and Deployment:** No analysis was conducted on computational cost for enterprise-scale deployment or integration complexity with existing compliance infrastructure. Future work should address these practical concerns, including performance benchmarking, resource requirements, and compatibility with regulatory reporting standards.

## 7 CONCLUSION

We presented a novel hybrid framework that enhances the explainability and verifiability of Large Language Models by transforming unstructured outputs into structured Feature Trees and merging them into a unified Uber Tree. Our approach combines LLM-based semantic abstraction with traditional algorithmic merging for robustness, complemented by automated consistency verification to ensure reliability and provide clear audit trails.

Extensive evaluation demonstrates that our method reliably generates high-quality structures, produces coherent and abstract knowledge representations, and achieves strong consistency with held-out knowledge. Human evaluation with domain experts confirms that our structured explanations significantly outperform standard Chain-of-Thought reasoning, particularly in ease of verification for real-world compliance auditing.

This work provides a concrete pathway toward building transparent, accountable, and auditable AI systems. By offering structured and verifiable knowledge representations, our framework bridges the gap between LLM capabilities and the rigorous demands of high-stakes applications. We are releasing our code and data to facilitate further research in this critical area.

LARGE LANGUAGE MODEL USAGE

In accordance with the ICLR 2026 policy on LLM usage, we disclose the following:

Large Language Models (GPT-4.1 (OpenAI, 2023)) were used as a tool in the preparation of this manuscript. Their use was strictly assistive and did not constitute a contribution to the intellectual content of the research itself. The precise roles were:

- **Ideation & Brainstorming:** LLMs were used in the early stages to brainstorm potential names for the proposed framework (e.g., "Feature Tree," "Uber Tree") and to generate a preliminary list of evaluation metrics. All outputs were critically reviewed and heavily modified by the authors; the core research idea and methodological design are our own.
- **Writing Assistance:** LLMs were used for text polishing, such as rephrasing sentences for clarity and conciseness, correcting grammar, and ensuring a consistent academic tone across the manuscript. The models were never used to generate entire paragraphs or sections of core technical content (e.g., the methodology, results, analysis) de novo.
- **Code Assistance:** LLMs were used for writing boilerplate code and utilities (e.g., for parsing JSON outputs, managing API calls) to accelerate implementation. The core algorithmic logic (e.g., the recursive merge fallback, evaluation scripts) was designed and implemented by the authors.

The authors take full responsibility for the entire content of this paper, including the accuracy of all facts, results, and citations. The LLM was used as a tool under our direct supervision and control at all times, and all of its outputs were critically evaluated and edited by the human authors.

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

# A  APPENDIX

## A.1  SAMPLE Q&A PAIR

### A.1.1  SAMPLE MORTGAGE COMPLIANCE Q&A

```
Question: Can you guarantee me the lowest interest rate on the
↪  market right now?
Answer: While we strive to offer competitive rates, we can't
↪  guarantee the lowest interest rate on the market. Interest
↪  rates can vary based on a variety of factors, including your
↪  credit profile, loan type, and market conditions. \n\nYour
↪  loan officer will work with you to find the best possible rate
↪  for your specific situation. If you have any questions about
↪  specifics, your loan officer will walk you through later as a
↪  part of the mortgage process.
```

## A.2  PROMPT TEMPLATES

### A.2.1  FEATURE TREE GENERATION PROMPT

```
"""You are an expert at analyzing mortgage compliance Q&A. For the
↪  following question and answer, generate a hierarchical feature
↪  tree in JSON format that explains the answer structure. The
↪  tree must include exactly these fields:
{{
  "Root": "Question category (Rates/Fees/Approval/etc)",
  "KeyFactors": ["list", "of", "factors"],
  "Process": ["list", "of", "steps"],
  "Limitations": ["list", "of", "caveats"],
```

```
594      "NextSteps": ["list", "of", "actions"],
595      "Resources": ["list", "of", "links"]
596   }}
597
598   Important rules:
599   1. Only return valid JSON – no other text
600   2. All fields must be included
601   3. Arrays can be empty but must exist
602   4. Escape all quotes properly
603
604   Note that the objective of using this tree is to allow you to
605   ↪   logically explain how you arrived at your answer. This will
606   ↪   help human reviewers understand and structure insights or
607   ↪   provide explanations from the question and your answer. This
608   ↪   may be to explain why your answer makes sense and how you
609   ↪   arrived at it. Or it may be reviewing a structural steps to
610   ↪   explain to an external auditor.
611
612   Here's the Q&A to analyze:
613   Question: {question}
614   Answer: {answer}
615   """
```

A.2.2  UBER TREE MERGING PROMPT

```
"""You are an expert at merging hierarchical tree structures for
↪   mortgage compliance knowledge.
I have {len(trees)} individual trees that need to be merged into a
↪   single highly abstract and organized tree.

Each tree has a structure like:
{{
  "node": "node_name",
  "children": [
    {{"node": "child1", "children": []}},
    {{"node": "child2", "children": []}}
  ]
}}

Here are all the trees in JSON format:
{trees_json}

Please merge these trees intelligently by:
1. Creating broad, abstract categories that represent major
↪   domains of mortgage compliance (aim for 8-12 top-level
↪   categories)
2. Grouping similar nodes under appropriate high-level parent
↪   categories (e.g., "Credit Factors", "Loan Terms", "Fees &
↪   Costs", "Regulatory Compliance")
3. Maintaining a clean hierarchical structure with meaningful
↪   abstractions
4. Removing duplicates while preserving unique information
5. Focusing on high-level concepts rather than granular details
6. Ensuring the tree is well-organized and easy to navigate

Examples of abstract top-level categories:
- Rates & Pricing
- Loan Products & Eligibility
- Application & Approval Process
- Fees & Closing Costs
```

```
648    - Documentation & Verification
649    - Regulatory Compliance
650    - Customer Service & Support
651    - Resources & Tools
652
653    Return ONLY the merged tree as a valid JSON object with the exact
654    ↪   same structure:
655    {{
656      "node": "Mortgage Compliance Knowledge Base",
657      "children": [
658        // 8-12 abstract top-level categories, each with appropriate
659        ↪   subcategories
660      ]
661    }}
662
663    Do not include any additional text, explanations, or markdown
664    ↪   formatting. Just the JSON object.
665    """
666
667
```

### A.2.3 CONSISTENCY VERIFICATION PROMPT

```
668    You are an expert in mortgage compliance knowledge structures.
669    I have an Uber tree that represents the comprehensive taxonomy of
670    ↪   mortgage compliance knowledge:
671    {uber_tree_json}
672
673    And I have an individual feature tree that needs to be verified
674    ↪   for consistency:
675    {test_tree_json}
676
677    Please analyze if this individual tree is consistent with the Uber
678    ↪   tree. Specifically:
679    1. Check if all nodes in the individual tree can be logically
680    ↪   mapped to the categories in the Uber tree.
681    2. Identify any nodes that do not fit into the Uber tree's
682    ↪   structure or are outliers.
683    3. Note if there are any missing categories in the Uber tree that
684    ↪   should be added based on this individual tree.
685    4. Provide a consistency score from 0 to 1, where 1 means
686    ↪   perfectly consistent.
687
688    Return your response in JSON format with the following structure:
689    {
690      "consistent": boolean,
691      "score": float,
692      "issues": list of strings describing inconsistencies,
693      "missing_categories": list of strings suggesting missing
694      ↪   categories,
695      "verdict": string summary
696    }
```

### A.3 CODE IMPLEMENTATION DETAILS

Our implementation consists of three main Python scripts:

**generate_feature_trees.py** - Uses OpenAI GPT-4.1-2025-04-14 with temperature=1.0 - Implements retry mechanisms for API failures - Processes Q&A pairs into structured JSON trees - Outputs to mortgage_compliance_answers_dmx_with_trees.jsonl

**build_uber_tree_llm.py**  - Implements chunked processing (10 trees per API call) - Uses GPT-4.1-2025-04-14 with temperature=0.1 for consistent abstraction - Includes fallback to traditional recursive merging - Outputs comprehensive Uber tree in JSON format

**evaluate_uber_tree.py**  - Performs 90/10 train-test split for evaluation - Uses GPT-4.1 for automated consistency verification - Generates structured audit reports with scores and issues - Outputs verification results to JSON file

## A.4   SAMPLE TREE STRUCTURES

### A.4.1   SAMPLE FEATURE TREE

```
{
  "Root": "Rates & Pricing",
  "KeyFactors": [
    "Credit score and history",
    "Loan-to-value ratio",
    "Debt-to-income ratio",
    "Market conditions",
    "Loan type and term"
  ],
  "Process": [
    "Check current market rates",
    "Evaluate borrower profile",
    "Calculate personalized rate",
    "Discuss options with loan officer"
  ],
  "Limitations": [
    "Rates subject to change",
    "No guarantee of lowest rate",
    "Individual eligibility varies"
  ],
  "NextSteps": [
    "Consult with loan officer",
    "Provide documentation",
    "Compare multiple offers"
  ],
  "Resources": [
    "https://owning.com/mortgage-rates/",
    "https://www.consumerfinance.gov/"
  ]
}
```

