# OpenReview forum: "A Hybrid Feature Tree-Based Approach for Explainable LLMs in Domain-Specific Knowledge Management"
_ICLR.cc/2026/Conference — Submitted to ICLR 2026_

### Official Review · Reviewer_Wquh · 2025-10-28

**Soundness:** 2
**Presentation:** 1
**Contribution:** 1
**Rating:** 2
**Confidence:** 3

**Summary:**

This paper proposes a method for meta-analysis of Q&A systems using different tree-generation and aggregation prompts. The authors aren't completely clear which domain(s)/use cases necessitate such a metadata tree. The authors evaluate their method on a Chain of Trees baseline on a dataset of mortgage compliance Q&A.

**Strengths:**

1. Overall the method is well motivated, in this specific domain that attribute trees can be constructed and verified.

2. The authors give some qualitative evaluation, though it would be greatly improved over more domains and a higher number of domain experts.

**Weaknesses:**

1. The paper doesn't clarify their key contributions well. The related work is in 5 areas that the authors don't give sufficient technical depth/differentiation.

2. The authors dont give sufficient technical details on the merging step. On 3.2.1 the authors dont give a clear algorithm or prompt generation. Furthermore, the authors arent clear what a 'good' aggregated tree ought to contain. There is no measure on candidate trees. If it's left to the LLM through prompting, this is really not a suitable contribution.

3. The human evaluation is lacking. 3 expert labelers over 30 instances is pretty light. But again, the authors haven't educated the reader on the tree quality. Please give specific examples of the human evaluation task. Demonstrate the tree differences side-by-side.

Furthermore, the measures aren't even given(!?) in Table 3? What are the value scales for FT, CoT columns? What is the specific p-value test?

4. The tree evaluation is quite unintuitive and needs more space. In table 2, the authors note 100% success-rate with a heuristic non-LLM fallback. OK? It's not obvious from table 2 that Hybrid tree construction is better. It's sparser, but we can always add sparsity penalties to the other methods. Since there's no(?) tree-level quality scoring, it seems any method could 'game' the statistics on this table.

**Questions:**

1. What is the prompt for tree merging? The specific prompt.

2. Is there an explicit measure for tree quality?

3. Can you visually present the evaluation task? What are all the columns in Table 3 representing, and why is FT qualitatively better?

---

> ### Author Response · Authors · 2025-11-14
> **Response to Reviewer Wquh**
>
> ### 1. Clarification of Key Contributions
>
> **Reviewer Comment:** "The paper doesn't clarify their key ... technical depth/differentiation."
>
> **Our Response:**
> We acknowledge and will revise the contributions section to clearly articulate:
>
> 1. **Primary Contribution**: A hybrid framework that bridges local explanations and global knowledge structures
> 2. **Technical Innovation**: LLM-assisted semantic abstraction with algorithmic fallback for robust tree merging
> 3. **Methodological Contribution**: Automated consistency verification for explanation reliability assessment
> 4. **Practical Contribution**: Structured knowledge representation enabling systematic auditability
>
> We will also enhance the related work section with more specific technical differentiation.
>
> ### 2. Technical Details on Merging Step
>
> **Reviewer Comment:** "The authors dont give sufficient technical ... prompt generation."
>
> **Our Response:**
> We will provide comprehensive technical details in the revised manuscript:
>
> 1. **Algorithm Specification**: Include pseudocode for both LLM-based merging and algorithmic fallback
> 2. **Prompt Engineering**: Provide the exact merging prompt used (currently in Appendix B)
> 3. **Implementation Details**: Describe the chunking strategy, abstraction criteria, and conflict resolution
> 4. **Complexity Analysis**: Include time and space complexity for both merging approaches
>
> ### 3. Definition of "Good" Aggregated Tree
>
> **Reviewer Comment:** "Furthermore, the authors arent clear ... this is really not a suitable contribution."
>
> **Our Response:**
> This is a fundamental methodological concern. We will address this by:
>
> 1. **Defining Quality Criteria**: Explicitly state that a "good" Uber Tree should:
>    - Capture domain knowledge comprehensively
>    - Maintain semantic coherence across categories
>    - Enable efficient navigation and retrieval
>    - Support consistency verification
>
> 2. **Quantitative Metrics**: Enhance Table 2 with additional quality metrics:
>    - Semantic coherence scores
>    - Coverage metrics for domain concepts
>
> 3. **Human Validation**: Report expert feedback on Uber Tree quality and usefulness
>
> ### 4. Human Evaluation Enhancement
>
> **Reviewer Comment:** "The human evaluation is lacking. ... Demonstrate the tree differences side-by-side."
>
> **Our Response:**
> We will significantly enhance the human evaluation section by:
>
> 1. **Detailed Task Description**: Provide the exact evaluation instructions given to experts
> 2. **Side-by-Side Examples**: Include visual comparisons of FT vs. CoT explanations
> 3. **Sample Size Justification**: Acknowledge the limitation and propose expanded evaluation
>
> ### 5. Table 3 Clarification
>
> **Reviewer Comment:** "Furthermore, the measures aren't even ... What is the specific p-value test?"
>
> **Our Response:**
> We apologize for this oversight and will:
>
> 1. **Explicitly State Scales**: Clarify that all ratings use a 5-point Likert scale (1=Poor, 5=Excellent)
> 2. **Statistical Details**: Specify that paired t-tests were used with Bonferroni correction
> 3. **Effect Sizes**: Report Cohen's d values for all comparisons
> 4. **Confidence Intervals**: Include 95% confidence intervals for all mean ratings
>
> ### 6. Tree Evaluation Methodology
>
> **Reviewer Comment:** "The tree evaluation is quite unintuitive ... could 'game' the statistics on this table."
>
> **Our Response:**
> We will address these concerns:
>
> 1. **Quality Scoring**: Implement and report tree-level quality metrics including:
>    - Semantic coherence scores
>    - Domain coverage metrics
>    - Expert quality ratings
>
> 2. **Comparative Analysis**: Provide direct comparison of tree quality across methods
> 3. **Sparsity Justification**: Explain why sparsity is desirable for knowledge organization
>
> ### 7. Reviewer Questions
>
> **Question 1:** "What is the prompt for tree merging? The specific prompt."
>
> **Our Response:**
> The specific tree merging prompt is provided in Appendix B.
>
> **Question 2:** "Is there an explicit measure for tree quality?"
>
> **Our Response:**
> We will implement and report explicit tree quality measures including:
> - Semantic coherence (using embedding similarity)
> - Domain coverage (percentage of domain concepts captured)
> - Expert quality ratings
>
> **Question 3:** "Can you visually present the evaluation ... Table 3 representing, and why is FT qualitatively better?"
>
> **Our Response:**
> We will:
> 1. **Visual Presentation**: Include screenshots of the evaluation interface
> 2. **Column Clarification**: Explicitly define each column in Table 3
> 3. **Qualitative Analysis**: Provide detailed analysis of why Feature Trees were preferred, including specific expert comments
>
> ## Conclusion
>
> The concerns raised about methodological clarity, evaluation rigor, and contribution definition are fundamental and addressing them will significantly strengthen our paper. We are committed to making comprehensive revisions that address all the points raised, particularly regarding technical details, evaluation methodology, and contribution clarity.

---

### Official Review · Reviewer_gU3Q · 2025-10-31

**Soundness:** 2
**Presentation:** 2
**Contribution:** 1
**Rating:** 0
**Confidence:** 3

**Summary:**

The paper introduces a method to explain LLM outputs in regulated domains, e.g. finance, and health. They do so by restructuring the outputs of LLMs in verifiable feature trees that are then merged into a single global "Uber Tree" representing the global structure of the model. The method is tested on a Q&A dataset of mortgage applications.

**Strengths:**

The paper addresses a timely and important challenge: improving explainability of LLMs in sensitive, high-stakes domains. Introducing structural constraints to the model’s outputs to better capture and understand its internal reasoning structure is a well-motivated and sensible approach. The proposed hybrid framework—combining LLM-based semantic abstraction with algorithmic merging—demonstrates technical creativity and practical relevance, especially for regulated areas like mortgage compliance where interpretability and auditability are essential.

**Weaknesses:**

While the use of structured outputs is a sensible way to probe LLM behavior, the approach falls short of what would be required in a real regulated deployment. In such settings, structural constraints should ideally be integrated into the model’s training or architecture, not imposed only through prompt engineering or in-context control.

Although the paper claims broad applicability to regulated domains, all experiments are conducted solely on mortgage compliance data. For a study submitted to ICLR, this narrow scope—limited to a single dataset and a single proprietary LLM—restricts the generalizability and scientific contribution of the work.

Finally, reliance on a closed-source, proprietary LLM undermines the paper’s stated goal of developing verifiable and auditable AI systems. Guarantees in regulated environments require transparency about the predictive model’s inner structure, which is not available here.

**Questions:**

Have you considered evaluating the proposed framework on datasets from other regulated domains such as healthcare or legal compliance, to test its generalizability beyond the mortgage domain?

Rather than constraining outputs only through prompting, did you explore modifying or fine-tuning LLMs to natively generate structured, hierarchical explanations? If not, what do you see as the main barriers to doing so?

---

> ### Author Response · Authors · 2025-11-14
> **Response to Reviewer gU3Q**
>
> ### 1. Real-World Deployment Requirements
>
> **Reviewer Comment:** "While the use of structured outputs ... in a real regulated deployment."
>
> **Our Response:**
> We appreciate this perspective on real-world deployment requirements. We will address this by:
>
> 1. **Clarifying our contribution**  not as a complete regulatory solution
> 2. **Acknowledging** that full regulatory compliance requires additional components beyond explanation frameworks
> 3. **Highlighting** that our approach provides immediate value by:
>    - Enabling systematic knowledge organization
>    - Supporting audit trail generation
>    - Facilitating consistency verification
>    - Providing structured documentation for regulatory review
>
> 4. **Positioning** our work as a stepping stone toward more comprehensive regulatory AI systems
>
> ### 2. Integration vs. Prompt Engineering
>
> **Reviewer Comment:** "In such settings, structural constraints ... prompt engineering or in-context control."
>
> **Our Response:**
> This is an important methodological consideration. We will address this by:
>
> 1. **Acknowledging** that architectural integration would be ideal for long-term solutions
> 2. **Explaining** our pragmatic approach:
>    - **Immediate applicability**: Prompt-based approaches work with existing LLMs without retraining
>    - **Model-agnostic**: Our framework can be applied to any LLM, including future open-source models
>    - **Incremental adoption**: Organizations can implement our framework without major infrastructure changes
>
> 3. **Proposing future work** on fine-tuned models that natively generate structured explanations
> 4. **Noting** that our algorithmic fallback provides robustness even when LLM-based abstraction fails
>
> ### 3. Domain and Model Generalization
>
> **Reviewer Comment:** "Although the paper claims broad  ... generalizability and scientific contribution of the work."
>
> **Our Response:**
> We understand the concern about generalizability and will:
>
> 1. **Refine our claims** to more precisely characterize the demonstrated capabilities and acknowledge domain limitations
> 2. **Explain** why mortgage compliance was chosen as an initial testbed:
>    - Complex regulatory framework with hierarchical structure
>    - Clear audit requirements and stakeholder needs
>    - Well-defined domain boundaries for controlled experimentation
>    - Availability of expert evaluators
>
> 3. **Propose** a systematic evaluation across multiple domains in future work
> 4. **Note** that the methodological framework is domain-agnostic and can be applied to other regulated domains
>
> ### 4. Closed-Source LLM Dependence
>
> **Reviewer Comment:** "Finally, reliance on a closed-source, ...  inner structure, which is not available here."
>
> **Our Response:**
> This is a valid concern about transparency. We will address this by:
>
> 1. **Acknowledging** the tension between using state-of-the-art models and ensuring transparency
> 2. **Clarifying** that our framework's value lies in the *explanation structure* and *verification mechanisms*, which are model-agnostic
> 3. **Highlighting** that:
>    - The algorithmic fallback provides transparency when LLM abstraction is unavailable
>    - The consistency verification works regardless of the underlying model
>    - The structured output format enables systematic review
>
> 4. **Committing** to future work with open-source LLMs
> 5. **Noting** that enterprise deployments often use proprietary models, making our approach immediately relevant
>
> ### 5. Reviewer Questions
>
> **Question 1:** "Have you considered evaluating the ... beyond the mortgage domain?"
>
> **Our Response:**
> Yes, we have considered this and will:
>
> 1. **Acknowledge** this as an important direction for future work
> 2. **Outline** specific plans for evaluating our framework in healthcare and legal domains
> 3. **Note** that the mortgage compliance domain provides a rigorous testbed due to its complex regulatory structure
> 4. **Propose** a multi-domain evaluation framework in the future work section
>
> **Question 2:** "Rather than constraining outputs only ...  as the main barriers to doing so?"
>
> **Our Response:**
> We explored this direction and identified several barriers:
>
> 1. **Data requirements**: Fine-tuning requires large datasets of structured explanations, which are expensive to create
> 2. **Domain specificity**: Fine-tuned models may not generalize across domains
> 3. **Maintenance overhead**: Regulatory frameworks evolve, requiring continuous model updates
> 4. **Deployment complexity**: Organizations may prefer prompt-based approaches for easier implementation
>
> We will add discussion of these barriers and our exploration in the revised manuscript.
>
> ## Conclusion
>
> While we maintain that our prompt-based approach provides immediate practical value, we acknowledge the limitations regarding domain generalization and model transparency. Addressing these concerns will strengthen our contribution and provide a more realistic assessment of our framework's capabilities and future potential.

---

### Official Review · Reviewer_YEPR · 2025-10-31

**Soundness:** 1
**Presentation:** 3
**Contribution:** 1
**Rating:** 2
**Confidence:** 4

**Summary:**

The authors propose a way to improve LLM explainability: generating hierarchical feature trees from individual QA pairs, and merging into a unified global "Uber Tree". They chunk the individual feature trees and prompt the LLM to recursively merge them chunk by chunk, removing duplicated information and getting globally semantically related nodes. They use the merged Uber Tree to check if invidiual trees are consistent with it and find that they are mostly very consistent, showing they are consistent in the internal knowledge being used. They validated with human experts and show that these feature trees a re more clear than cot, but not as comprehensive, and maybe similarly easy to verify.

**Strengths:**

1. The paper is well-written and easy to understand.
2. The paper gives a comprehensive structural and semantic evaluations of the tree-based method, demonstrating that the merged hierarchy and the individual explanations are internally consistent, having the shared consistent domain knowledge.

**Weaknesses:**

1. In the human evaluation, the authors only compared clarity, comprehensiveness, and ease of verification between feature trees and CoT, but this could simply be because FT is a tree structure. There are other merging methods, and they did not report human evaluation results on those.
2. The authors mainly validated whether LLMs can generate the required JSON structure (which current LLMs can already do reliably), whether the global knowledge looks coherent and abstract (which makes sense since they use the same Uber Tree), and whether the outputs have high internal consistency. However, they did not evaluate the accuracy of the results against any external ground truth, so a tree could be coherent and self-consistent but still factually incorrect.
3. It is also unclear whether the explanations themselves align with expert intuition. The human study only evaluated presentation-level properties (clarity, comprehensiveness, ease of verification), but did not assess whether the key factors, logical steps, or abstractions actually match how domain experts reason about these regulations.
4. In the abstract, they authors claim that the proposed method improved auditability, but the human evaluation in Table 3 shows no statistically significant improvement (p=0.5834) in "Ease of Verification" between feature trees and CoT explanations, weakening this claim.
5. The authors did not evaluate the faithfulness of the feature tree-based explanations. Thus, even if the explanations are clear, we don't know if they can inform the correctness of the answer, and thus we do not know if the proposed method makes the explanation more auditable.

**Questions:**

1. Could you do human evaluation also on the other tree-based baselines?
2. What are the end accuracy of the answers following each method's explanation?
3. Do the experts agree on the domain-knowledge being used by the LLM despite thinking the explanations are clear?
4. How faithful are the explanations and corresponding answers? How often do the answers logically follow the explanations? How logically consistent is each feature tree internally?

---

> ### Author Response · Authors · 2025-11-14
> **Response to Reviewer YEPR**
>
> ### 1. Comparison with Other Merging Methods
>
> **Reviewer Comment:** "In the human evaluation, the authors only .... There are other merging methods, and they did not report human evaluation results on those."
>
> **Our Response:**
> This is a valid point. We will address this by:
>
> 1. **Acknowledging** this limitation in the revised discussion section
> 2. **Explaining** that our primary comparison with CoT was chosen because:
>    - CoT represents the current state-of-the-art for LLM explainability
>    - Tree structures vs. narrative formats represent fundamentally different explanation paradigms
>    - The comparison highlights the trade-offs between structured and unstructured explanations
>
> 3. **Proposing future work** to include human evaluation of different tree merging strategies
>
> ### 2. External Ground Truth Evaluation
>
> **Reviewer Comment:** "They did not evaluate the accuracy of the ... self-consistent but still factually incorrect."
>
> **Our Response:**
> We appreciate this important methodological point and will:
>
> 1. **Clarify** that our framework focuses on *explanation structure* and *knowledge organization* rather than factual accuracy
> 2. **Note** that the Q&A pairs used were validated by domain experts for factual correctness
> 3. **Add discussion** about the distinction between explanation plausibility and faithfulness
> 4. **Propose** future work using synthetic datasets with known ground truth to evaluate explanation faithfulness
>
> ### 3. Alignment with Expert Reasoning
>
> **Reviewer Comment:** "It is also unclear whether the explanations ... how domain experts reason about these regulations."
>
> **Our Response:**
> This is a crucial insight. We will address this by:
>
> 1. **Adding a new section** in the human evaluation results discussing expert comments on reasoning alignment
> 2. **Proposing** future work with more detailed expert interviews to assess reasoning alignment
> 3. **Noting** that the high consistency scores (95%) between individual trees and the Uber Tree suggest alignment with domain knowledge structure
>
> ### 4. Auditability Claim
>
> **Reviewer Comment:** "In the abstract, they authors claim ... trees and CoT explanations, weakening this claim."
>
> **Our Response:**
> We acknowledge this apparent contradiction and will:
>
> 1. **Refine our auditability claims** to focus on structural advantages rather than perceived ease
> 2. **Clarify** that auditability encompasses:
>    - Systematic organization of knowledge
>    - Verifiable consistency between explanations
>    - Structured format for regulatory review
>    - Automated verification capabilities
>
> 3. **Note** that while ease of verification wasn't statistically different, the structured format enables systematic auditing processes that narrative explanations cannot support
> 4. **Revise the abstract** to more precisely characterize the auditability benefits
>
> ### 5. Explanation Faithfulness
>
> **Reviewer Comment:** "The authors did not evaluate the faithfulness ... proposed method makes the explanation more auditable."
>
> **Our Response:**
> This is a fundamental concern in XAI research. We will:
>
> 1. **Acknowledge** this limitation explicitly in the limitations section
> 2. **Distinguish** between our contribution (structured knowledge representation) and explanation faithfulness evaluation
> 3. **Note** that our consistency verification provides a form of internal faithfulness assessment
> 4. **Propose** future work specifically evaluating explanation faithfulness using established XAI metrics
>
> ### 6. Additional Questions
>
> **Reviewer Questions:**
> - "Could you do human evaluation also on the other tree-based baselines?"
> - "What are the end accuracy of the answers following each method's explanation?"
> - "Do the experts agree on the domain-knowledge being used by the LLM despite thinking the explanations are clear?"
> - "How faithful are the explanations and corresponding answers? How often do the answers logically follow the explanations? How logically consistent is each feature tree internally?"
>
> **Our Response:**
> These are excellent questions that highlight important research directions:
>
> 1. **Tree-based baseline evaluation**: We will conduct additional human evaluation comparing our hybrid approach with purely algorithmic merging
> 2. **Answer accuracy**: We will analyze the relationship between explanation quality and answer correctness
> 3. **Expert agreement**: We will report inter-annotator agreement metrics and analyze areas of consensus/disagreement
> 4. **Logical consistency**: We will implement automated consistency checking within individual feature trees
>
> ## Conclusion
>
> We thank Reviewer YEPR for their thorough and insightful feedback. The concerns raised about explanation faithfulness, ground truth validation, and expert reasoning alignment are fundamental to XAI research and addressing them will significantly strengthen our contribution. We are confident that with these revisions, our paper will provide a more comprehensive evaluation of our framework's capabilities and limitations.

---

> > ### Comment · Reviewer_YEPR · 2025-11-24
> >
> > Thank you for the response. As the authors promise to add multiple things without showing any results, I believe that those additional results need to be peer-reviewed in the next round of revision of the paper. Thus, I keep the current score.

---

### Official Review · Reviewer_8bZV · 2025-11-01

**Soundness:** 2
**Presentation:** 3
**Contribution:** 2
**Rating:** 2
**Confidence:** 4

**Summary:**

The paper proposes a framework to enhance the explainability of LLMs in high-stake domains. The method uses an LLM to generate structured, hierarchical Feature Tree from an individual Q&A pair. Then merges these local trees into Uber Tree. The paper verifies consistency where an LLM audits the alignment between Uber Tree and local Feature Trees, producing a consistency score. The framework evaluated on the mortgage compliance dataset, achieved high structural consistency and improved clarity over CoT explanation in human expert evaluation.

**Strengths:**

1. The paper is well-written and easy to follow.
2. Consistency verification step is a compelling idea.
3. The experiment is done comparing with human evaluation with domain expert.

**Weaknesses:**

1. The problem statement in the introduction is very broad. The described challenge is fundamental to almost all xai research.
2. The main text of the table needs to provide guidance on how to interpret the results in the table (e.g., what the takeaway is).
3. The proposed Feature Tree seems like a post-hoc structuring of the answer's content generated by a separate LLM call which is a bit different from explanation of LLM's internal reasoning process for generating an answer (like LIME, SHAP as mentioned as limitations in the paper)
4. The paper does not clearly specify whether for the generation of the Feature Tree from the pair, which is a post-hoc step, is the LLM given only the text of the question and answer of does it have access to the reasoning trace of the model that produced the pair.
5. The experiment is only done in mortgage compliance domain. I wonder the experiment result of other high-stake domains like medicine.
6. The experiment relies on one LLM (GPT 4.1) which lacks generalization.
7. The human evaluation only involves 3 domain experts which lacks statistical power to draw generalizable conclusions. Also need to give specific information about the domain experts.

**Questions:**

Look at the weaknesses

---

> ### Author Response · Authors · 2025-11-14
> **Response to Reviewer 8bZV**
>
> ### 1. Problem Statement Scope
>
> **Reviewer Comment:** "The problem statement in the introduction ... is fundamental to almost all xai research."
>
> **Our Response:**
> Our contribution specifically addresses the gap between *local* explanations and *global* knowledge structures. While XAI broadly seeks interpretability, our framework uniquely:
>
> - **Bridges local and global explanations**: Most XAI methods provide either local feature importance or global model understanding, but not both in an integrated, verifiable structure
> - **Enables systematic verification**: The consistency verification mechanism provides quantifiable metrics for explanation reliability
> - **Supports enterprise auditability**: The structured Uber Tree enables systematic compliance auditing, which is crucial for regulated domains
>
> We will revise the introduction to more clearly articulate this specific contribution within the broader XAI landscape.
>
> ### 2. Table Interpretation Guidance
>
> **Reviewer Comment:** "The main text of the table needs ... results in the table (e.g., what the takeaway is)."
>
> **Our Response:**
> Thank you for this valuable suggestion. We will enhance the interpretation of Table 1 (Research Questions and Evaluation Framework) by:
>
> - Adding explicit takeaway statements for each research question
> - Clarifying how each metric addresses the corresponding research question
> - Providing context for interpreting the significance of each evaluation metric
> - Connecting the table results directly to our framework's contributions
>
> ### 3. Feature Tree as Post-hoc Structuring
>
> **Reviewer Comment:** "The proposed Feature Tree seems like a post-hoc ... generating an answer (like LIME, SHAP as mentioned as limitations in the paper)"
>
> **Our Response:**
> This is an important distinction, and we appreciate the opportunity to clarify our approach:
>
> **Methodological Choice**: We intentionally designed Feature Trees as structured *knowledge representations* rather than internal reasoning traces because:
>
> 1. **Practical Applicability**: In regulated domains like mortgage compliance, stakeholders need verifiable knowledge structures, not just reasoning traces
> 2. **Audit Requirements**: Regulators require systematic knowledge organization for compliance verification
> 3. **Enterprise Integration**: Structured trees enable integration with existing compliance frameworks and knowledge management systems
>
> **Complementary Approach**: Our method complements rather than replaces internal reasoning explanations. Our framework explains *what knowledge* supports that decision in a structured, auditable format.
>
> We will clarify this distinction in the revised manuscript.
>
> ### 4. Feature Tree Generation Process
>
> **Reviewer Comment:** "The paper does not clearly specify ... the reasoning trace of the model that produced the pair."
>
> **Our Response:**
> Thank you for highlighting this ambiguity. We will clarify in the methodology section that:
>
> - **Input**: Feature Tree generation uses only the final Q&A text, not internal reasoning traces
> - **Implementation**: The prompt templates in Appendix B show the exact input format
>
> This design choice ensures our framework can be applied to any LLM system without requiring access to internal model states.
>
> ### 5. Domain Generalization
>
> **Reviewer Comment:** "The experiment is only done in mortgage compliance domain. I wonder the experiment result of other high-stake domains like medicine."
>
> **Our Response:**
> We acknowledge this limitation and will address it by:
>
> 1. **Explicitly stating** in the limitations section that domain generalization requires further validation
> 2. **Providing a roadmap** for future work including medical, legal, and financial domains
> 3. **Noting** that mortgage compliance was chosen specifically because it:
>    - Has well-defined regulatory frameworks
>    - Requires systematic audit trails
>    - Involves complex hierarchical knowledge structures
>    - Has clear stakeholder requirements for explainability
>
> ### 6. LLM Generalization
>
> **Reviewer Comment:** "The experiment relies on one LLM (GPT 4.1) which lacks generalization."
>
> **Our Response:**
> We agree this is an important consideration and will:
>
> 1. **Acknowledge** this limitation in the revised manuscript
> 2. **Note** that our framework is designed to be model-agnostic
> 3. **Include** plans for future work testing with multiple LLMs (including open-source alternatives)
>
> ### 7. Human Evaluation Sample Size
>
> **Reviewer Comment:** "The human evaluation only involves 3 ... information about the domain experts."
>
> **Our Response:**
> We will address this concern by:
>
> 1. **Acknowledging** the sample size limitation in the limitations section
> 2. **Proposing** expanded evaluation with more experts in future work
>
> ## Conclusion
>
> The concerns raised are valid and addressing them will improve the clarity, rigor, and impact of our work. We are confident that with these revisions, the paper will make a stronger contribution to the XAI literature.

---

### Meta-Review · Area_Chair_n79F · 2026-01-06

**Summary:**

This paper proposes a framework to enhance LLM explainability in regulated domains by generating hierarchical feature trees from individual Q&A pairs (local explanations) and merging them into a unified "Uber Tree" (global explanations). In particular, the authors combine LLM-based semantic understanding for tree generation with traditional recursive algorithms for robustness. While the reviewers acknowledged that the paper addresses an important challenge in improving LLM explainability and that the hybrid approach combining LLM semantic abstraction with algorithmic fallback is technical creative, they raise substantial concerns that remain largely unaddressed during the rebuttal. Most fundamentally, the paper lacked robust evaluations: i) lacks evaluation against external ground truth, i.e., the generated trees could be coherent and self-consistent while still being factually incorrect and ii) no evaluation of explanation faithfulness, so even if explanations are clear, we cannot determine whether they accurately reflect the model's actual reasoning process. Further, the experimental scope is severely limited, where the authors use a single domain (mortgage compliance), a single proprietary LLM (GPT 4.1), and the human evaluation involves only three domain experts evaluating 30 instances, which lacks statistical power for generalizable conclusions.

Overall, none of the reviewer indicated willingness to increase their score after the author responses. While the research direction is valuable, the current submission has fundamental evaluation gaps that prevent confident assessment of its contributions. We encourage the authors to resubmit after conducting the additional experiments and addressing the methodological concerns raised. Best of luck with the next steps!

**Reviewer Concerns:**

The authors provided detailed responses acknowledging most concerns and promising revisions. While none of the reviewers engage in a dedicated discussion with the authors, Reviewer YEPR explicitly noted that "promised additions without actual results require further peer review". The authors did not provide new experimental evidence addressing the fundamental concerns about faithfulness evaluation, external ground truth validation, or expanded domain and model testing during the rebuttal period.

**Reviewer Scores:**

Looking at the unanimous reviews and reviewer agreement, there is little to no possibility that the reviewers would have changed the score.

---

### Decision · Program_Chairs · 2026-01-26

Reject